# Geometric and Morphometric Analysis of the Auditory Ossicles in the Red Fox (*Vulpes vulpes*)

**DOI:** 10.3390/ani13071230

**Published:** 2023-04-01

**Authors:** Nedžad Hadžiomerović, Ozan Gundemir, Faruk Tandir, Rizah Avdić, Muhamed Katica

**Affiliations:** 1Department of Basic Sciences of Veterinary Medicine, Veterinary Faculty, University of Sarajevo, 71000 Sarajevo, Bosnia and Herzegovina; 2Department of Anatomy, Faculty of Veterinary Medicine, Istanbul University-Cerrahpasa, Istanbul 34500, Turkey; 3Department of Clinical Sciences of Veterinary Medicine, Veterinary Faculty, University of Sarajevo, 71000 Sarajevo, Bosnia and Herzegovina

**Keywords:** ear ossicles, geometric morphometrics, morphometry, red fox

## Abstract

**Simple Summary:**

The aim of the study was to determine the morphological and morphometrical features of the auditory ossicles in the red fox. Moreover, for the first time, shape analysis was performed on all the middle ear bones by geometric morphometry. The auditory ossicles play a vital role in transmitting sound waves through the middle ear. The malleus was considered to be the largest bone, with three distinctive processes. The overall shape of the incus was found to be similar to that in other mammals. The presence of a prominent muscular process was noted on the head of the stapes and the convex base which closes the window of the inner ear.

**Abstract:**

The use of carnivores as experimental models in auditory biology has led to a significant improvement regarding our understanding of the structure and function of the ear. Considering that data regarding the anatomy of the middle ear in the red fox are rare, this study aimed to describe the morphological and morphometrical features of the auditory ossicles in the red fox, as well as to provide their shape characteristics by geometric analysis. Nine adult red foxes were used in the study. The malleus, incus and stapes were extracted from the middle ear, prepared, photographed and measured using the software. For the geometric analysis, 19 landmarks were used. Following Principal Component Analysis (PCA), PC1 was found to explain approximately half of all variance (incus: 49.97%; malleus: 49.93%; stapes: 58.49%). The study demonstrated the similar anatomical organization of the auditory ossicles in line with important morphometric and basic geometric data, which can contribute to this field and add a useful perspective to the literature.

## 1. Introduction

The extremely adaptable and most numerous representative of the *Vulpes* genus is the red fox (*Vulpes vulpes* Linnaeus, 1758). It is present in the steppe biomes and mixed forests of Eurasia within the temperate climate zone [1,2]. The red fox has adapted to life in different types of habitats and climates and can be seen in urban and suburban areas, which has led to partial changes in behavior and in its manner and type of nutrition [1,3]. *Vulpes vulpes* belongs to the order of Carnivora; however, unlike dogs and cats, it is actually an undomesticated canid, between a large cat and a small dog in size [4]. However, as the largest species of the genus *Vulpes*, its body size ranges between 3 and 14 kg [5]. The ear (organum vestibulocochleare) is a complex sense organ that includes both the organs of balance and of hearing. As the organ is subdivided into three parts, sound waves are received by the external ear (Auris externa), transmitted through the middle ear (Auris media), and transformed into electrical signals by the inner ear (Auris interna), where the vestibular organ is located [6]. The middle ear extends from the tympanum (membrana tympani) to the oval vestibular window (fenestra vestibuli) and houses three auditory ossicles joined in the form of a chain. In many mammals, the tympanic cavity extends ventrally from the skull base and forms the bulla tympanica [7]. The middle ear cavity is connected with the pharynx by way of the auditory or Eustachian tube. The ossicles mediate transmission of vibrations through the tympanic cavity to the inner ear [8,9]. The most lateral and largest bone is the malleus, which consists of a head, neck and handle, embedded in the tympanic membrane. The incus is divided into a body with two limbs and a lenticular process. The stapes consist of a head, two limbs and a base [5,10]. One of the best studied species in auditory biology is the domestic cat (*Felis sylvestris catus*). The use of this species as an experimental model has led to a great deal being known about the structure and function of the ear [4,11,12,13]. The auditory biology and morphometry of the ears of dogs (*Canis lupus familiaris*) have also been relatively well studied [4,14]. The anatomical features of the middle ear and its parameters are often used to predict the hearing sensitivity of the species. Although the red fox has similar anatomical structures of the middle ear to dogs and cats, the morphological aspect of ear structures should be taken with caution when considering the hearing attributes of related species [4]. Furthermore, the anatomical and morphological characteristics of the hearing apparatus of the red fox have been described, and it was determined that the middle ear is very similar to that of the dog; however, the cochlear structures of red foxes are very similar to those of cats [4]. Detailed descriptions of the morphological features and morphometric parameters of the bulla tympanica, tympanic membrane and middle ear ossicles have been composed for the red fox [15]. The study revealed values for several parameters of the auditory ossicles, as well as weight parameters and area measurements for the tympanic membrane, and the oval and round window. Similar morphological studies on the middle ear structures were performed on carnivores, such as wolves [10], dogs [16,17], badgers [18], red foxes, dogs and cats [4], on domestic and wild carnivores [19], on some herbivores, such as goats [20], sheep [21], horses [22], donkeys [23,24], on ruminants [25], on rodents, such as mice [26], hamsters [27], rabbits [9,28], chinchillas [29,30], and on some birds, such as ostriches [31] and chickens [32]. Geometric morphometry provides more precise information about small anatomical structures and focuses on data analysis from the position of homologous landmarks [33,34]. The process of analysis excludes factors such as position and size, revealing only shape-related differences [35]. To the best of the authors’ knowledge, no data were found about the detailed morphometry of the auditory ossicles and their anatomical parts in the red fox. The present study aims to describe detailed morphological and morphometrical data regarding the auditory ossicles of the red fox, as well as to reveal their basic shape characteristics using the geometric morphometric method. 

## 2. Materials and Methods

### 2.1. Study Sample

Nine red foxes (seven females and two males) were used in this study. All the animals were adults according to their teeth formula, aged between 6–12 months. Additional methods for age determination were the fusion of the cranial sutures and cementum layers on the longitudinal section of the canine tooth. The crania were all obtained from the Department of Pathology after autopsies were performed. The muscles and the fascia were dissected, boiled and macerated to remove fat, and finally the skulls were bleached with 3% hydrogen peroxide. After the cleaning procedure was complete, the ossicula auditus were extracted from the tympanic cavity via the external auditory channel.

### 2.2. Morphometric–Analysis

The samples were prepared and digital images were collected using a stereomicroscope (ZEISS Stemi 508) under 0.63 magnification with a camera (ZEISS Axiocam 208, Jena, Germany). The images were further processed using ZEISS ZEN 3.0 software (blue edition) and measured using ImageJ^®^ software after the calibration procedure. The samples were measured in order to determine the differences between the auditory ossicles on the right and left sides.

The parameters for measurement of the auditory ossicles were selected according to previous studies [10,18,29].

The following measurements were performed (Figure 1):LM—length of the malleusWHM—width of the head of the malleusLHM—length of the head of the malleusLhM—length of the handle of the malleusLI—length of the incusLLC—length of long crus of the incusLSC—length of short crus of the incusHBI—height of the body of the incusWBI—width of the body of the incusLS—length of the stapesLCC—length of the caudal crus of the stapesLRC—length of the rostral crus of the stapesWHS—width of the head of the stapesWBS—width of the base of the stapes

The anatomical structures of the auditory ossicles were identified and named according to the NAV [36] and other relevant literature [18,19].

### 2.3. Geometric Analysis

For the geometric morphometric analysis, digital images were taken with the same magnification using the stereomicroscope. The files were converted into “tps” format using the tpsUtil (version 1.74) program [37]. The landmarks used in the study are shown in Figure 2. TpsDig2 (version 2.32) was used for landmarking.

Incus, 1—leftmost peak of the crus longum; 2—rightmost peak of the crus longum; 3—angle of the crus breve with the crus longum; 4—highest point of the crus breve; 5—left corner point of corpus incudis; 6—middle point of the articulation incundomalleolaris;

Malleus, 7—connecting point of the caput mallei and collum mallei; 8—top of processus rostralis; 9—top of the processus muscularis; 10—connecting of the manubrium mallei; 11—terminal point of the manubrium mallei; 12—upper segment of the processus lateralis; 13—lower segment of the processus lateralis;

Stapes, 14—highest connecting point of the caput and crus caudale; 15—highest connecting point of the caput and crus rostrale; 16—connecting point of the crus caudale and basis stapedis; 17—caudal point of the basis stapedis; 18—connecting point of the crus rostrale and basis stapedis: 19—rostral point of the basis stapedis.

### 2.4. Statistical Analysis

MorphoJ (version 1.07a) software was used for the statistical part of the geometric morphometric analysis. The landmark file was imported into MorphoJ, and “Procrustes fit” was applied first. A Generalized Procrustes Analysis was applied to the imported landmark data before analysis. Principal Component Analysis (PCA) was performed to determine the shape variations among the bones.

## 3. Results

The auditory ossicles of the red fox are composed of three bones: the malleus, incus and stapes. The malleus, as the most external lateral part of the ossicles chain, is connected to the tympanic membrane and articulates with the middle ossicle, the incus, followed by the most internal auditory ossicle, the stapes, which closes the fenestra vestibuli. The largest bone is the malleus (Figure 3), consisting of three distinctive anatomical parts: the head of the malleus (caput mallei), the neck of the malleus (collum mallei) and the handle (manubrium mallei). The head of the malleus is slightly ovoidal in shape and connected with the neck. The articular surface, in the form of a saddle-like impression, is located on the caudo-medial aspect of the head. The head (caput) is laterally divided from the neck and osseous lamina by a small notch. These three processes were identified on the malleus of the red fox. The anterior process (processus rostralis) is located near the head and is triangular in shape, oriented laterally. The longer muscular process (processus muscularis) originates near the base of the manubrium and has a sharp ending. The muscular process extends ventrally in the same direction as the manubrium and attaches to the m. tensor tympani muscle. Opposite the muscular process is the short, lateral process (processus lateralis). The handle of the malleus represents around 70% of the total malleus length. The total malleus length (LM) measures 7.21 ± 0.53 mm on the right and 7.18 ± 0.53 mm on the left side, while the manubrium (LhM) measures on the right 5.19 ± 0.53 mm and left 5.27 ± 0.42 mm, respectively (Table 1). At the beginning, the manubrium has three sides in cross section; however, further along distally, it continues as a double-sided structure. The terminal part of the manubrium is embedded in the tympanic membrane.

The incus, the second auditory ossicle, is significantly smaller than the malleus, measuring 2.34 ± 0.38 mm on the right side and 2.25 ± 0.09 mm on the left side. The overall shape of the incus resembles a biradicular molar tooth (Figure 4). Situated in the middle, it is connected with both auditory ossicles. With its body (corpus incudis) it articulates with the malleus, while its longer crus (crus longum) at its terminal part forms an articular surface with the lenticular process. This process ossifies with the long crus in later life, while juvenile animals have a separate additional bone, the lenticular bone (os lenticulare).

The articular surface of the corpus is divided by a notch into two smaller parts. The short crus (crus breve) continues in the dorso-caudal direction, ending in a pointed shape. The size of the two limbs is similar, but the right side is slightly larger (Table 1).

The stapes is the third and smallest auditory ossicle (Figure 5). As the innermost ossicle, it is interposed between the incus and the vestibular window, completing the transmission of vibrations to the perilymph in the inner ear. The total length of the stapes is the same on both sides, 2.11 ± 0.10 mm. The stapes is triangular in shape, consisting of the head (caput stapedis), two limbs, the rostral (crus rostrale), the posterior (crus caudale) and the base (basis stapedis). The head is a spherical structure that forms the articulation with the lenticular process. The small prominent part of the head represents the muscular process (tuberositas m. stapedius), used as the insertion point of the stapedial muscle (m. stapedius). This muscle, together with the malleus-attached muscle (m. tensor tympani), has an important role in the vibration enhancement mechanism in the inner ear. The head is distally connected with the base by the rostral and caudal crus, which are quite similar in size (Table 1). The small opening in the form of a tunnel (foramen intercrurale) between the limbs and base is closed by a stapedial membrane (membrana stapedis). The base of the stapes has an elliptical shape with central convexity which closes the window (fenestra vestibuli). This connection is secured with the annular ligament. 

PCA was performed to reveal the inter-individual shape variations of the auditory ossicles. PC results for each bone are given in Table 2. For each bone, PC1 was found to explain approximately half of all variance (incus: 49.97%; malleus: 49.93%; stapes: 58.49%). PCA results and shape variations are given in Figure 6, Figure 7 and Figure 8.

Wire-frame warp plots of changes in the malleus shape in PC1, PC2 and PC3 are shown in Figure 5. Compared to PC1, the most variation in the shape of the malleus was seen in the collum mallei. The higher PC2 value represents the larger caput mallei. In PC3, the positive value represents the longer processus muscularis. Shape variations in the incus were generally in the middle part of the corpus incudis. There were also inter-individual shape variations in the processus lenticularis. In the stapes, inter-individual shape variations were generally seen in the foramen intercrurale (Figure 8). There were differences in shape between individuals at the borders of the foramen intercrurale.

The mean shape for female and male is given in Figure 9. In incus, the highest point of the crus breve in males was more oval in shape than in females. In addition, the angle of the crus breve with the crus longum in males was narrower than in females. The tip of the male manubrium mallei was wider in shape than that of the females. The connecting point of the caput mallei and collum mallei was deeper in females. Additionally, the intercrural foramen in the stapes was wider in shape in females.

## 4. Discussion

The morphological characteristics of the auditory structures are closely associated with different environmental and behavioral patterns [13,38]. Some reports have suggested, at least in principle, that hearing ability can be predicted from the anatomical structures [39]. However, despite the significant difference in body and tympanic membrane size, behavioral audiograms are similar in various breeds of dog [40]. The red fox audiogram showed similar hearing characteristics with both dogs and cats [41]. The middle ear structures have been relatively well studied in domestic carnivores [12,13,14]. The middle ear apparatus of the fox, including the auditory ossicles, is, as expected, very similar to that of cats and dogs [4]. The present study revealed a detailed morphometric analysis of the auditory ossicles of the red fox, including shape analysis, using geometric morphometry.

The three auditory ossicles are located in the middle ear of the red fox, as in most mammals: the malleus, incus and stapes. The macroanatomical features of the malleus revealed a similar shape to that in wolves [10]. The overall size of the malleus in the fox was slightly smaller than that of the badger and goat but larger than that of the cat, hamster and macaque monkey [18,19,27,42]. The ovoidal shape of the head of the malleus in the fox was similar to that found in previous studies in carnivores [10,18,19]. The muscular and rostral processes on the malleus were longer than the lateral process in the wolf [10], which is similar to our findings in the fox. The size of the anterior and shorter lateral processes of the malleus was 2.25 ± 0.25 mm and 1.49 ± 0.11 mm, respectively [15]. The robust and curved manubrium was typical for carnivores [4], while its triangular shape in cross-section was in accordance with the results reported from the badger and goat [18,20]. 

Previous reports described the shape of the incus as similar to a biradicular molar tooth and also as an anvil with a saddle [4,9,15,18,20]. The two divergent crura differ greatly in size in the majority of carnivores and other mammals [9,18,29]. However, our results reported very small differences between the longer and short crus of the incus (Table 1). Slightly smaller values of both limbs of the incus were reported in a previous study in the red fox [15]. Similar sizes of the two incus limbs were reported in goats and water buffaloes [20,43]. 

The smallest auditory ossicle, the stapes, has a unique triangular shape, similar in many species, and it is composed of the head, two crura and the base of the stapes [9,10,29]. The specific rectangular shape of the stapes has been noted in ruminants [25]. The red fox stapes was recorded as most fragile of the three ossicles with a close resemblance to a stirrup [15]. The small head on the top carries the prominent muscular process for the stapedial muscle, which was previously described in rabbits, dogs and goats [9,18,20]. The overall length of the stapes was smaller than in the wolf [10], badger [18] and goat [20]. A previous report on the length of the stapes in the red fox showed identical results to ours, and the study was performed on a larger number of individuals (n = 26) compared to our study [15]. The rostral crus of the stapes was slightly longer than the caudal crus; however, no differences were found between the right and left middle ears. A longer caudal crus of the stapes has been described in goats, hamsters and chinchilla [20,27,29]. The base of the stapes was represented by a little oval plate which was connected to the vestibular window through an annular ligament [25]. The study measured the weight parameters for ear ossicles in red fox; the mean weight of the malleus was 10.64 ± 1.18 mg, for incus 5.19 ± 0.63 mg and for stapes 0.63 ± 0.13 mg [15].

Recent studies have used geometric morphometric methods for various species to contribute to the taxonomic classification of animals and to determine sex dimorphism [35,44,45,46,47]. One study examined the shape variations of the incus and the differences between the donkey and the horse. The results reported that the corpus incudis edges (right, left, and bottom) were flatter in donkeys [48]. The study revealed that taxonomy was possible for ear ossicles using the geometric analysis method. Sex discrimination within the same species may be possible using geometric shape analysis [44]. The differentiation was more evident in the dorsal aspect of the quail skull. In our results, shape variations between the male and female were close to each other. Some differences were noted in part of the collum of the malleus and the intercrural foramen in the stapes. The study was carried out with a small sample for sexual dimorphism, so further advances in this field depend very much on the availability of specimens of both sexes and different age groups.

## 5. Conclusions

The study presented a morphological description of the auditory ossicles in the red fox, alongside detailed morphometrical data of the particular parts of the ossicles. The results showed that the anatomical organization of the auditory ossicles was similar to that of other carnivores, in terms of morphometric parameters, which may contribute to the study of the comparative morphology of the middle ear. The study revealed the further potential of shape analysis of the auditory bones using the geometric morphometry method, which should provide a significant addition to traditional morphometric studies. However, some limitations should be considered regarding this study and its results. Geometric analysis should aim to include a larger sample size in order to obtain more details about the shape differences between males and females. Moreover, it would be useful to use more than one aspect of the bones, as the previous study showed. Overall, we believe that further geometric morphometric studies of the ear ossicles would be very useful and would contribute to the morphology of these bones.

## Figures and Tables

**Figure 1 animals-13-01230-f001:**
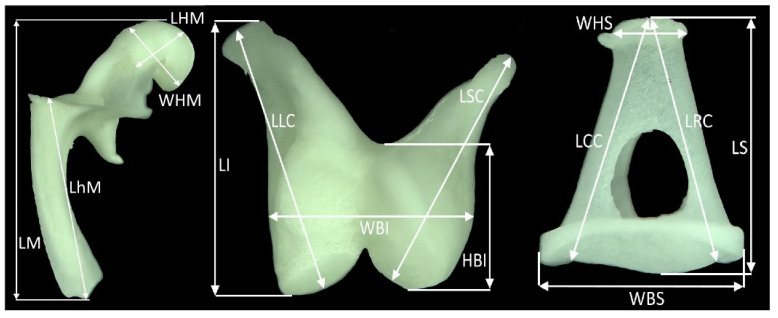
Measurements of the auditory ossicles. Abbreviations—see text.

**Figure 2 animals-13-01230-f002:**
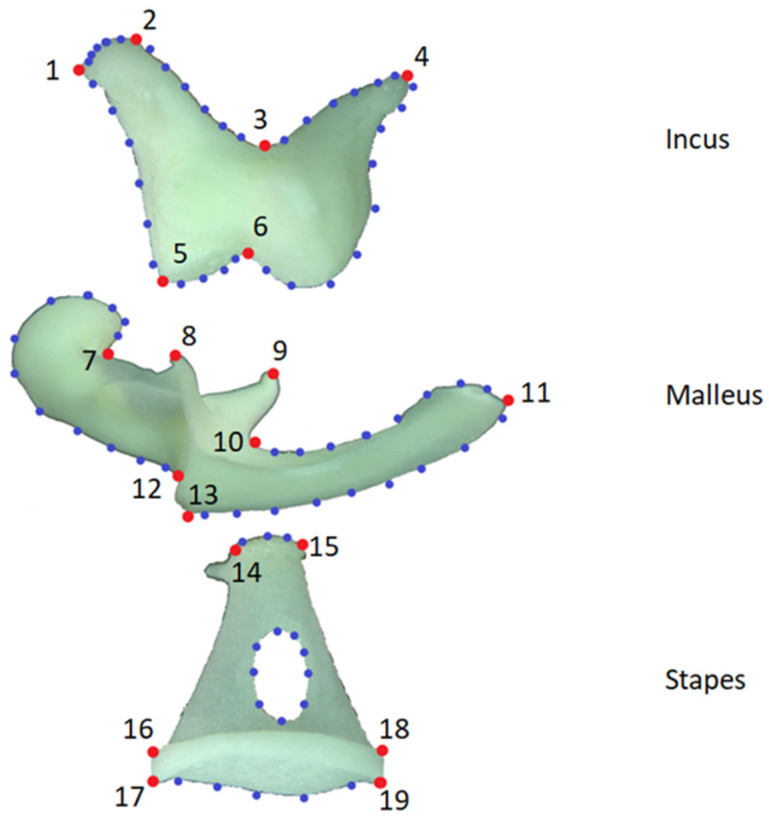
Landmarks (red) and Semilandmarks (blue).

**Figure 3 animals-13-01230-f003:**
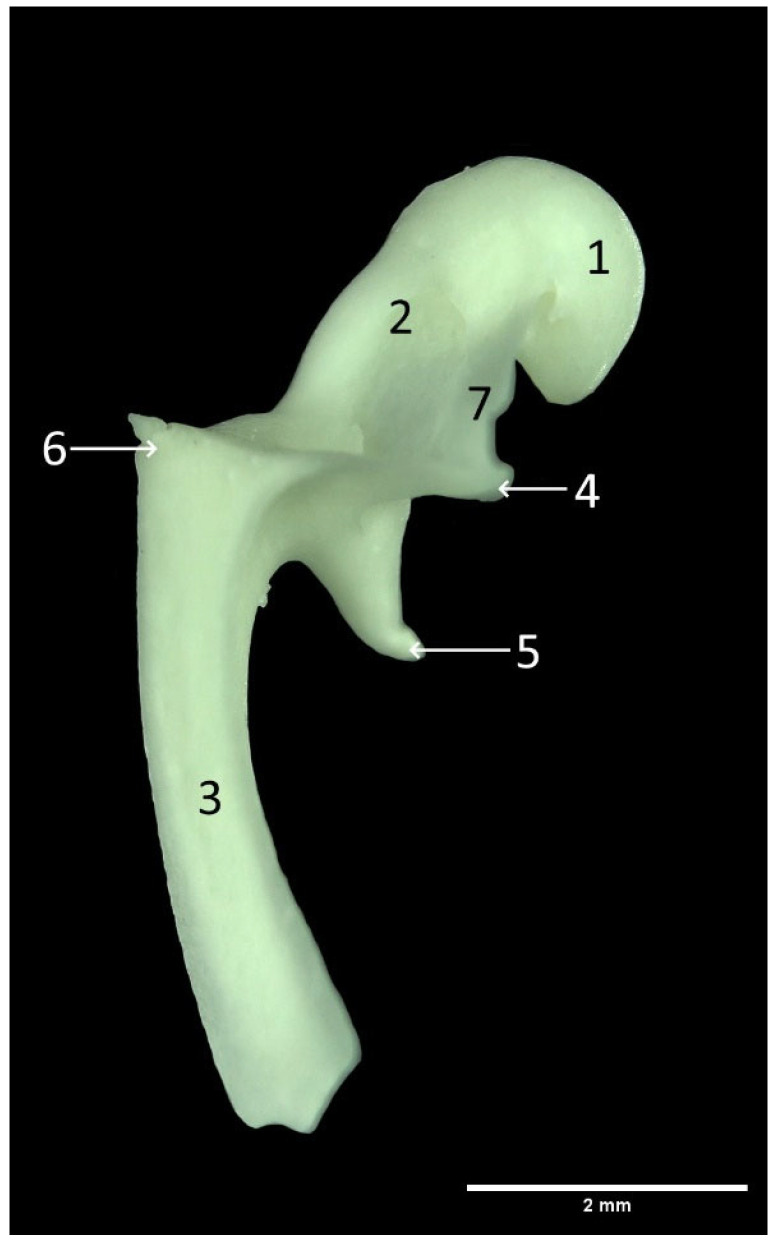
Malleus. 1. Caput mallei; 2. Collum mallei; 3. Manubrium mallei; 4. Processus rostralis; 5. Processus muscularis; 6. Processus lateralis; 7. Lamina.

**Figure 4 animals-13-01230-f004:**
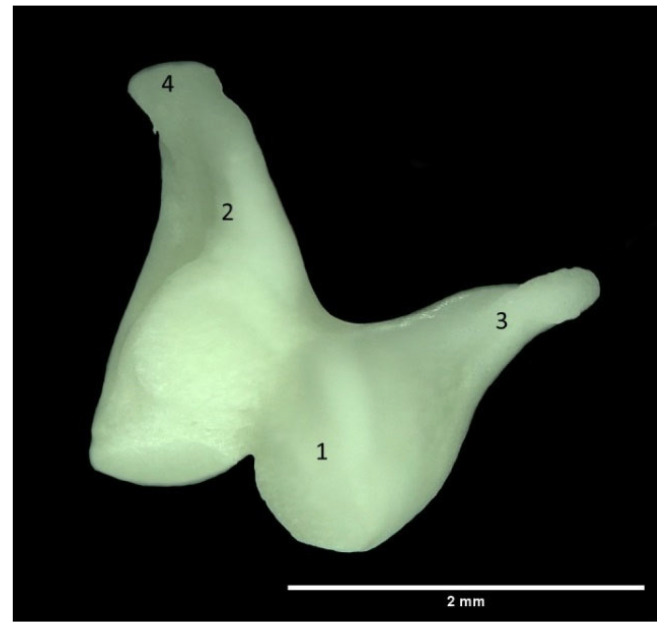
Incus. 1. Corpus incudis; 2. Crus longum; 3. Crus breve; 4. Processus lenticularis.

**Figure 5 animals-13-01230-f005:**
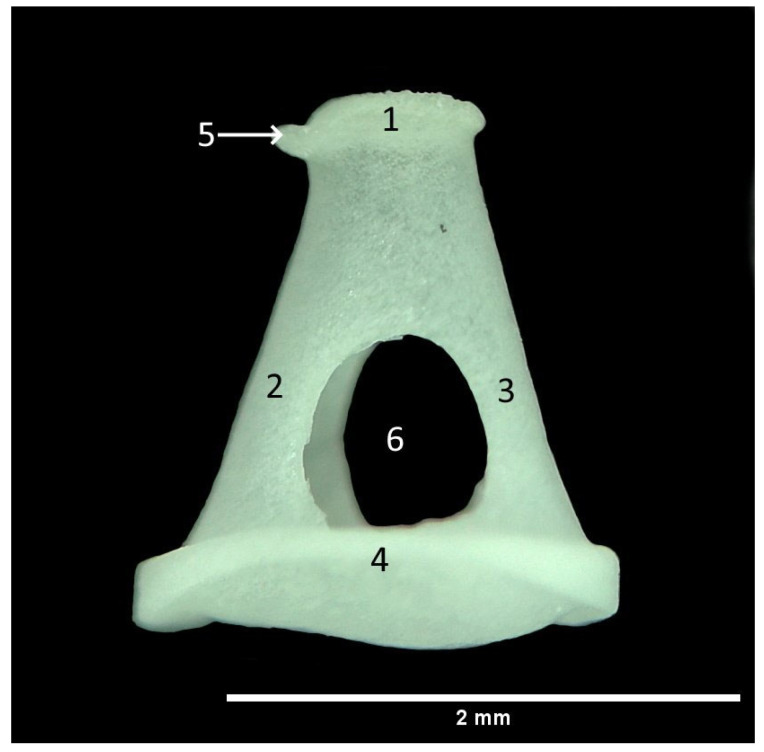
Stapes. 1. Caput stapedis; 2. Crus caudale; 3. Crus rostrale; 4. Basis stapedis; 5. Tuberositas m. stapedius; 6. Foramen intercrurale.

**Figure 6 animals-13-01230-f006:**
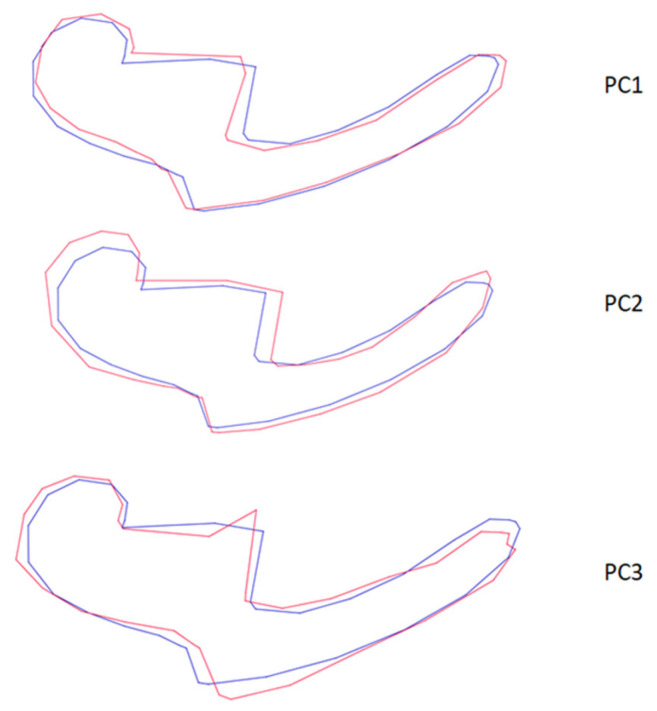
Wire-frame warp plots of changes in the malleus shape in PC1, PC2 and PC3. The blue plots indicate the initial shape, the red plots indicate the positive limit of the principal component’s scores.

**Figure 7 animals-13-01230-f007:**
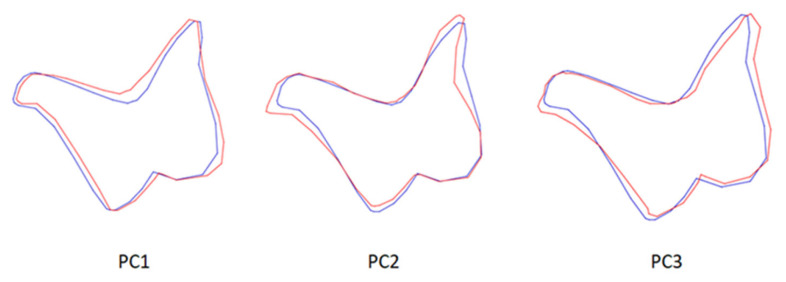
Wire-frame warp plots of changes in incus shape in PC1, PC2 and PC3. The blue plots indicate the initial shape; the red plots indicate the positive limit of the principal component’s scores.

**Figure 8 animals-13-01230-f008:**
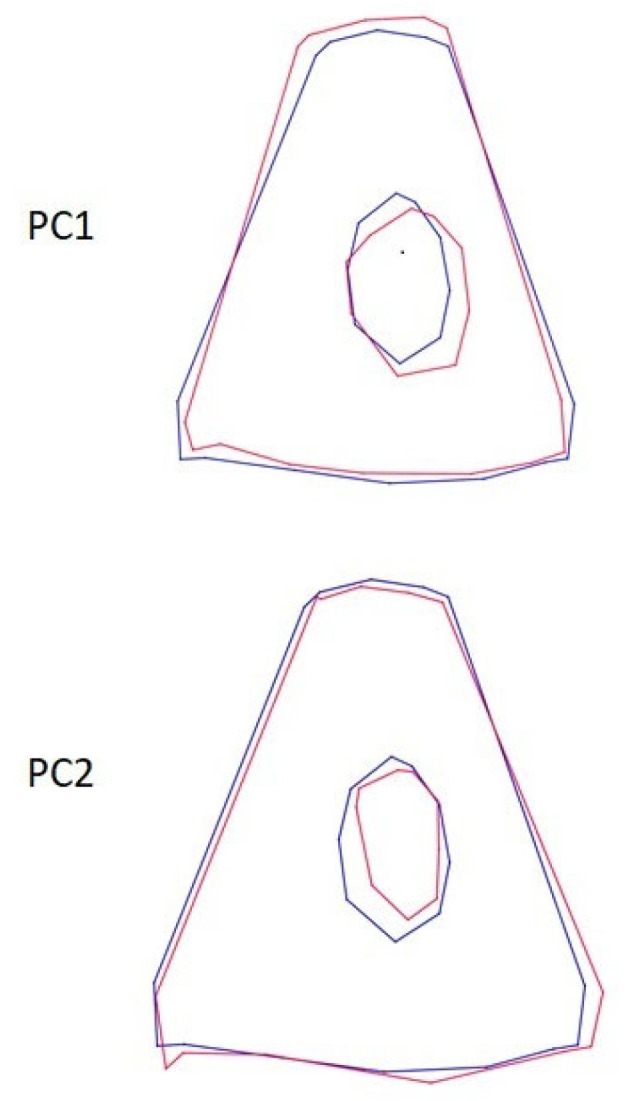
Wire-frame warp plots of changes in shape of the stapes in PC1 and PC2. The blue plots indicate the initial shape; the red plots indicate the positive limit of the principal component’s scores.

**Figure 9 animals-13-01230-f009:**
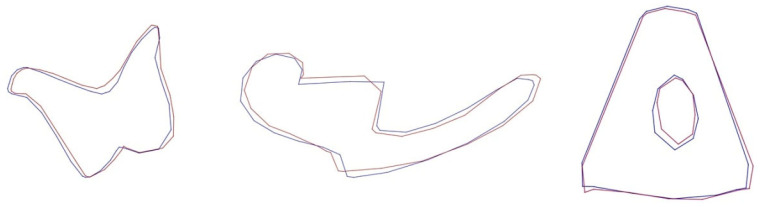
Main shapes for female (blue) and male (red).

**Table 1 animals-13-01230-t001:** Morphometric values of the auditory ossicles (mm).

Measurement	No	Right Side Mean ± SD	Left Side Mean ± SD	Red Fox (Malkemper, 2014) [15]	Badger (Martonos et al., 2022) [18]	Wolf—Rs (Gürbüz et al., 2019) [10]
**Malleus**						
LM	7	7.21 ± 0.53	7.18 ± 0.53		8.14 ± 0.14	9.35 ± 0.14
WHM	7	1.75 ± 0.07	1.73 ± 0.10		2.26 ± 0.014	2.16 ± 0.14
LHM	7	1.15 ± 0.10	1.19 ± 0.10			1.60 ± 0.12
LhM	7	5.19 ± 0.53	5.27 ± 0.42	3.22 ± 0.38 * 7.04 ± 0.31 **	4.85 ± 0.15	6.73 ± 0.67
**Incus**						
LI	7	2.34 ± 0.38	2.25 ± 0.09		2.72 ± 0.12	3.01 ± 0.32
LLC	7	2.44 ± 0.47	2.33 ± 0.11	2.06 ± 0.12	2.69 ± 0.21	3.09 ± 0.23
LSC	7	2.39 ± 0.54	2.23 ± 0.17	2.14 ± 0.19	1.99 ± 0.1	2.73 ± 0.65
HBI	7	1.31 ± 0.27	1.22 ± 0.08		1.48	1.73 ± 0.25
WBI	7	2.19 ± 0.41	2.01 ± 0.10		2.07 ± 0.14	2.26 ± 0.14
**Stapes**						
LS	3	2.11 ± 0.10	2.11 ± 0.10	2.11 ± 0.15	2.3 ± 0.08	2.57 ± 0.12
LCC	3	2.05 ± 0.14	2.07 ± 0.11		2.09 ± 0.2	2.51 ± 0.03
LRC	3	2.11 ± 0.10	2.11 ± 0.10		1.95 ± 0.11	2.77 ± 0.09
WHS	3	0.61 ± 0.05	0.61± 0.07		0.79 ± 0.13	0.49 ± 0.12
WBS	3	1.88 ± 0.10	1.86 ± 0.09		2.16	2.01 ± 0.21

* Measured as the distance from the presumed axis of rotation. ** Measured from the pivotal point of the malleo-incudal joint. LM—length of the malleus; WHM—width of the head of the malleus; LHM—length of the head of the malleus; LhM—length of the handle of the malleus; LI—length of the incus; LLC—length of long crus of the incus; LSC—length of short crus of the incus; HBI—height of the body of the incus; WBI—width of the body of the incus; LS—length of the stapes; LCC—length of the caudal crus of the stapes; LRC—length of the rostral crus of the stapes; WHS—width of the head of the stapes; WBS—width of the base of the stapes.

**Table 2 animals-13-01230-t002:** Results of PCA (Principal Component Analysis) for the auditory ossicles.

PC	Incus			Malleus			Stapes		
	E	% V	C%	E	% V	C%	E	% V	C%
PC1	0.00362179	49.946	49.946	0.00373181	49.931	49.931	0.00198681	58.485	58.485
PC2	0.00204002	28.133	78.078	0.00172400	23.067	72.998	0.00141034	41.515	100.000
PC3	0.00085646	11.811	89.889	0.00100988	13.512	86.510	-	-	-
PC4	0.00050179	6.920	96.809	0.00049064	6.565	93.075	-	-	-
PC5	0.00023140	3.191	100.000	0.00033112	4.430	97.505	-	-	-
PC6	-	-	-	0.00018645	2.495	100.000	-	-	-

## Data Availability

The data presented in this study are available on request from the corresponding author.

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
