# Peer review of "Geometric and Morphometric Analysis of the Auditory Ossicles in the Red Fox (Vulpes vulpes)"

_animals, 2023, doi:10.3390/ani13071230_

Round 1
Reviewer 1 Report
Review:
The authors present anatomical measurements of the middle ear ossicles of the red fox and perform a geometric analysis.
It has to be stated from the very beginning that most of the data presented in this manuscript is not novel. Several of the linear measurements of the ossicles have already been published in the PhD work of Malkemper in 2014 based on a much larger sample size (n>40 compared to n=9 in the current manuscript. The thesis can be found online (I am happy to provide a copy if not): https://www.semanticscholar.org/paper/The-Sensory-Biology-of-the-Red-Fox-Hearing%2C-Vision%2C-Malkemper/cd0540776e3eee353035c28ff1a393041f6880bc ). Still, additional measurements might confirm or be in dispute with published findings so they might yield interesting insights. They have to be put in context of the original measurements, however, meaning that the study should be cited in the introduction and discussed in comparison to the new findings in the present manuscript. Furthermore, the authors did not discuss the hearing abilities of the red fox in their manuscript which are published as well (Malkemper et al. 2015, hearing research). In fact, the authors did not describe at all which knowledge gap they are trying to fill with their research. Describing the hearing properties of the red fox might be a way to do this because it does not easily correlate with auditory anatomy so far and more measurements might shed additional light on this issue.
As to the geometric analysis, in the way it is currently presented it does not make sense. All it shows is how the bones between individuals differ and since the difference might also be due to technical errors arising from the placement under the microscope (the data is a 2D projection only), the value of this analysis is zero. If information on the age or sex of the individuals is available, comparing the geometrics between these might yield more interesting information (as some of the authors have done before, e.g. with quail skulls). If the information on the individuals is not available, I would strongly suggest to remove this analyis.
Other major comments:
Line 15: What do the authors mean by most developed bone?
Line 43: internal ear should read inner ear
Line 44: The membrane is called tympanum
End of the introduction: What is the question of this study? Why has it been conducted?
2.1: What was the age and sex of the individuals used?
Measured parameters: Please show a figure that depicts the measured parameters to make it easier to compare these with other studies.
Line 146: Always present the means +-SD and state if SD or SEM is shown at first mention
Table1: Presenting the measurements from the Phd thesis of Malkemper (2014) alongside your measurements would a scientifically strong statement.
Line 211: This is wrong! The behavioural audiograms of cats and dogs do differ and the fox is different from both species.
Conclusions: The described shape variances of the bones do not yield any scientific insight, the final sentence is massively overstating this.
Whole text: Carefully check for typos and grammatical errors, I found several at the first read.
In summary, I cannot recommend this manuscript for publication at the current stage. In case the editor thinks that the content is interesting enough to be published, the above-mentioned comments should be adressed first.
Author Response
Response to Reviewer 1.
We would like to thank to the reviewers for their valuable comments. We appreciate the inputs they provided in order to help improving the manuscript. Our responses are given in a point-by-point manner, addressing all their concerns. Changes in the manuscript are highlighted in yellow.
Question 1.
It has to be stated from the very beginning that most of the data presented in this manuscript is not novel. Several of the linear measurements of the ossicles have already been published in the PhD work of Malkemper in 2014 based on a much larger sample size (n>40 compared to n=9 in the current manuscript.
Answer 1.
We didn’t find this thesis before, but after your suggestion we added in introduction and discussion section. This reference provide very important details about several parameters as you stated, so we compare our results with the previous study of Malkemper.
Q2.
They have to be put in context of the original measurements, however, meaning that the study should be cited in the introduction and discussed in comparison to the new findings in the present manuscript.
A2.
Corrected and added.
Q3.
Furthermore, the authors did not discuss the hearing abilities of the red fox in their manuscript which are published as well (Malkemper et al. 2015, hearing research). In fact, the authors did not describe at all which knowledge gap they are trying to fill with their research.
A3.
Our study focused on the anatomical description and detail morphometry of the era ossicles in the red fox. The hearing abilities wasn’t part of our study although we mentioned that morphological aspect does affect the hearing abilities and cited some research about that. Regarding the which knowledge gap we try to fill, our study focus on the anatomical description and morphometry like the recent studies published:
- Morphological and Morphometrical Aspects of the Auditory Ossicles in the European Badger (Meles Meles), 2022.
- Morphological and morphometrical aspects of the auditory ossicles in goat (Capra hircus), 2021.
- Morphometric and macroanatomic examination of auditory ossicles in male wolves (Canis lupus), 2019.
- Morphological and morphometrical study of the auditory ossicles in chinchilla, 2019.
- Macroanatomic and Morphometric Study on Ossicula Auditus in Male Hemshin Sheep, 2019.
Q4.
If information on the age or sex of the individuals is available, comparing the geometrics between these might yield more interesting information (as some of the authors have done before, e.g. with quail skulls).
A4.
We include this information in the study and performed geometric analysis between sex.
Q5.
Line 15: What do the authors mean by most developed bone?
Line 43: internal ear should read inner ear
Line 44: The membrane is called tympanum
A5.
Corrected.
Q6.
What was the age and sex of the individuals used?
Measured parameters: Please show a figure that depicts the measured parameters to make it easier to compare these with other studies.
A6.
We added this information.
The figure is added.
Q7.
Line 146: Always present the means +-SD and state if SD or SEM is shown at first mention.
A7.
Corrected.
Q8.
Table1: Presenting the measurements from the Phd thesis of Malkemper (2014) alongside your measurements would a scientifically strong statement.
A8.
We presented the results from Malkemper in the Table 1. as well as for other carnivores, wolf and badger.
Q9.
Conclusions: The described shape variances of the bones do not yield any scientific insight, the final sentence is massively overstating this.
A9.
We corrected the conclusion and add limitation of the study.
Q10.
Whole text: Carefully check for typos and grammatical errors, I found several at the first read.
A10.
The manuscript was corrected by native speaker and professional translator and language reviser and we provide this certificate.
Reviewer 2 Report
The study presented a morphological description of the auditory ossicles in red fox with employing both linear measurements and geometric morphometric approach. I find the study really interesting as the first detailed description of the red fox ossicles size and shape.
Some minor recommendation could be given. First, it would be great to give more detail description of used specimens: farm/wild captured foxes, adult/subadult, male/female, and museum numbers if any. Second, PC plots will be useful to demonstrate distribution of the studied specimens in the morphospace.
In general, I would like recommend the manuscript for publication in the “Animals” Journal.
Author Response
Response to Reviewer 2.
We would like to thank to the reviewer for valuable comments. We appreciate the inputs they provided in order to help improving the manuscript.
We gave more details about specimens, gender and age. As for PC plots, the study was performed on the small sample as we noted as the limitation of the study, so the problem was to demonstrate distribution in the morphospace.
Round 2
Reviewer 1 Report
The authors made an effort to respond to my comments and questions that is appreciated. However, some of my concerns are not yet fully adressed and some new ones have come up. I, therefore, ask for some additional minor revisions before the paper might be acceptable.
General: A reader interested in the morphology of auditory ossicles in the red fox will be interested in its hearing biology. Therefore, reference to its hearing abilities should be made if only to let the readers know that these have been studied. Please include the respective citations.
Line 98-100: Why are the authors interested in differences between the L and R ears? Has such a difference ever been reported in any species? There is no mentioning of this in the introduction. Also, I could not find the results of this analysis.
Line 152: Replace “most developed” by “largest” as done in the abstract. Please check to do this consistenly throughout the manuscript.
Lines 298-304: Where can these discussed differences between the sexes be found in the results section? I could not find a geometric analysis between sexes in the results.
Line 207: “PCA was performed to reveal the shape variations of the auditory ossicles.” It is still unclear to me which variations have been tested here. Between L&R, between sexes, between individuals? Please provide more details and be precise.
Author Response
Response to Reviewer 1.
We would like to thank to the reviewer for his/her valuable comments. Our responses are given in a point-by-point manner, addressing all your concerns. Changes in the manuscript are highlighted as track changes.
Q1.
General: A reader interested in the morphology of auditory ossicles in the red fox will be interested in its hearing biology. Therefore, reference to its hearing abilities should be made if only to let the readers know that these have been studied. Please include the respective citations.
A1.
We included the reference from Malkemper et al. 2015.
Q2.
Line 98-100: Why are the authors interested in differences between the L and R ears? Has such a difference ever been reported in any species? There is no mentioning of this in the introduction. Also, I could not find the results of this analysis.
A2.
Some previous study from Gurbuz et al. 2019 in wolves and Kurtul et al. 2003 in the rabbit measured parameters of the ear ossicles in the both right and left side. Some minor changes were noticed. We use this model and results are presented in the Table 1. Also, some results are presented in the Results section. For the stapes results were almost identical between the sides but for malleus and incus, some smaller differences were noticed.
Q3.
Line 152: Replace “most developed” by “largest” as done in the abstract. Please check to do this consistenly throughout the manuscript.
A3.
Corrected.
Q4.
Lines 298-304: Where can these discussed differences between the sexes be found in the results section? I could not find a geometric analysis between sexes in the results.
A4.
We added this part in the results section.
Q5.
Line 207: “PCA was performed to reveal the shape variations of the auditory ossicles.” It is still unclear to me which variations have been tested here. Between L&R, between sexes, between individuals? Please provide more details and be precise.
A5.
Corrected